# Repaglinide Induces ATF6 Processing and Neuroprotection in Transgenic SOD1G93A Mice

**DOI:** 10.3390/ijms242115783

**Published:** 2023-10-30

**Authors:** Rafael Gonzalo-Gobernado, Laura Moreno-Martínez, Paz González, Xose Manuel Dopazo, Ana Cristina Calvo, Isabel Pidal-Ladrón de Guevara, Elisa Seisdedos, Rodrigo Díaz-Muñoz, Britt Mellström, Rosario Osta, José Ramón Naranjo

**Affiliations:** 1National Centre for Biotechnology (CNB), Consejo Superior de Investigaciones Científicas (CSIC), 28049 Madrid, Spain; rd.gonzalo@cnb.csic.es (R.G.-G.); pgperez@cnb.csic.es (P.G.); jmdopazo@cnb.csic.es (X.M.D.); isabel.pidal@cnb.csic.es (I.P.-L.d.G.); elisa.seisdedos.castro@gmail.com (E.S.); rodrigodimu@gmail.com (R.D.-M.); bmellstr@gmail.com (B.M.); 2Centro de Investigación Biomédica en Red Enfermedades Neurodegenerativas (CIBERNED), Instituto de Salud Carlos III, 28029 Madrid, Spain; lauramm@unizar.es (L.M.-M.); accalvo@unizar.es (A.C.C.); 3LAGENBIO, Faculty of Veterinary, University of Zaragoza, Miguel Servet 177, 50013 Zaragoza, Spain; 4Aragón Health Research Institute (IIS Aragón), Biomedical Research Centre of Aragón (CIBA), 50009 Zaragoza, Spain; 5AgriFood Institute of Aragon-IA2 (UNIZAR-CITA), 50013 Zaragoza, Spain

**Keywords:** ALS, repaglinide, DREAM, ATF6, SOD1, UPR, motoneurons, microglia, astroglia

## Abstract

The interaction of the activating transcription factor 6 (ATF6), a key effector of the unfolded protein response (UPR) in the endoplasmic reticulum, with the neuronal calcium sensor Downstream Regulatory Element Antagonist Modulator (DREAM) is a potential therapeutic target in neurodegeneration. Modulation of the ATF6–DREAM interaction with repaglinide (RP) induced neuroprotection in a model of Huntington’s disease. Amyotrophic lateral sclerosis (ALS) is a neurodegenerative disorder with no cure, characterized by the progressive loss of motoneurons resulting in muscle denervation, atrophy, paralysis, and death. The aim of this work was to investigate the potential therapeutic significance of DREAM as a target for intervention in ALS. We found that the expression of the DREAM protein was reduced in the spinal cord of SOD1G93A mice compared to wild-type littermates. RP treatment improved motor strength and reduced the expression of the ALS progression marker collagen type XIXα1 (*Col19α1* mRNA) in the quadriceps muscle in SOD1G93A mice. Moreover, treated SOD1G93A mice showed reduced motoneuron loss and glial activation and increased ATF6 processing in the spinal cord. These results indicate that the modulation of the DREAM–ATF6 interaction ameliorates ALS symptoms in SOD1G93A mice.

## 1. Introduction

The Downstream Regulatory Element Antagonist Modulator (DREAM) protein, also known as calsenilin or KChIP3, is a member of the potassium channel interacting protein (KChIP) family of neuronal Ca^2+^ sensors [1,2,3]. Encoded by four genes, KChIP1 to 4, KChIP proteins show distinct expression patterns in different organs as well as in different areas within the CNS [1,3,4] and have multiple functions [5,6,7], sometimes specific for the different family members (recently reviewed in [8,9]). In the case of DREAM/KChIP3, hereafter DREAM, specialized activities are achieved via specific protein–protein interactions (PPI) and the Ca^2+^-dependent binding to specific sites in the DNA [3,10]. Calcium binding to functional EF-hands in the DREAM protein triggers a conformational change that prevents binding to DRE sites in the DNA [3] and modifies its affinity for some interacting proteins, e.g., CREM, CREB, and GRK2, but not for others, e.g., presenilins and Kv4 channels [10]. Importantly, changes in DREAM conformation are also induced upon binding of small molecules, like arachidonic acid [11], glinides [12] and some diaryl urea derivatives [13], which affects the interacting properties of DREAM and modifies its protein–protein interactions [10]. In a previous work, we reported that protein levels of DREAM were reduced in Huntington’s disease (HD) patients as well as in HD mice and HD knock-in cells [14]. We proposed that endogenous DREAM silencing in HD may be part of an early endogenous neuroprotective mechanism since (i) induced DREAM haplodeficiency in R6/2 mice, a transgenic HD mouse model, delayed the onset of motor dysfunction, reduced striatal atrophy, and prolonged lifespan, (ii) DREAM overexpression has the opposite effect in R6/2 mice and sensitized HD knock-in cells to mitochondrial stress, and (iii) blocking DREAM activity with DREAM-interacting molecules like repaglinide (RP) or PC330 reduced disease symptoms and cell viability. The process involves the interaction between DREAM and activating transcription factor 6 (ATF6), one of the three endoplasmic reticulum (ER) sensors that activate the unfolded protein response (UPR) in response to ER stress [15]. Exposure to DREAM ligands reduces the DREAM–ATF6 interaction, which increases ATF6 processing and improves the UPR and the neuronal survival [14]. Whether this mechanism applies to other neurodegenerative diseases is currently not known.

Amyotrophic lateral sclerosis (ALS) is the third most common neurodegenerative disease, after Alzheimer’s disease and Parkinson disease. ALS is characterized by a progressive loss of motoneurons accompanied by muscle atrophy and paralysis, triggering the death of the patient in an average period of 3 to 5 years after the onset of symptoms. Although a small percentage of ALS cases have a genetic origin with a known mutation, the majority of cases, 90–95%, are of unknown cause [16]. The first mutated gene associated with ALS was the gene of the superoxide dismutase 1 (SOD1) enzyme, which allowed researchers to develop the SOD1G93A mouse model [17], one of the best characterized SOD1 transgenic mice. Numerous mechanisms involved in the pathogenesis of ALS have been described, including mitochondrial dysfunction, alterations in RNA function and abnormal protein processing, excitotoxicity, oxidative stress, neuroinflammation, and axonal transport damage. Despite this, there is currently no treatment capable of curing the disease, only palliative treatments that delay the progression of the disease by a few months. Therefore, there is a need to find a therapy capable of curing the disease or, at least, slowing its progression.

A previous study using a pan-KChIP antibody [18] indicated that changes in KChIP levels during disease progression in SOD1G93A mice may correlate with the astrocytic response and the excitotoxic death of spinal cord motoneurons. Here, using an antibody specific for DREAM/KChIP3 which does not cross react with other KChIP family members [19], we disclose a significant reduction in DREAM protein levels in the lumbar spinal cord. Moreover, we show that the administration of the DREAM interacting molecule RP to SOD1G93A mice improved motor strength, reduced the astrocytic and the microglial activation, and partially rescued motoneurons from death. The mechanism involves an activation of ATF6 processing in the spinal cord of transgenic mice. The present study highlights the potential therapeutic relevance of DREAM as a target for intervention in the context of ALS associated neurodegeneration.

## 2. Results

### 2.1. Expression of DREAM Is Significantly Reduced in the Lumbar Spinal Cord of SOD1G93A Mice

To define a potential role for DREAM in ALS neurodegeneration, we assessed the level of DREAM protein in extracts of lumbar spinal cord from SOD1G93A mice and their wild-type littermates. Using a specific DREAM antibody, we found that both monomeric and tetrameric forms of DREAM were significantly reduced in the lumbar spinal cord of symptomatic SOD1G93A mice (Figure 1A–D). The decrease in DREAM protein was reminiscent of that previously observed in the brain of R6/2 mice, a mouse model of Huntington’s disease (HD), where a further block of DREAM activity with RP ameliorated disease symptoms [14]. To investigate whether this endogenous silencing of DREAM expression in SOD1G93A mice has also a neuroprotective function, we looked for a positive effect of the administration of the DREAM binding molecule, RP, in these mice.

### 2.2. Repaglinide Treatment Delayed the Loss of Motor Strength in SOD1G93A Mice

Ten-week-old SOD1G93A transgenic mice, corresponding to an early symptomatic stage of the disease, and WT littermates were exposed to RP in the drinking water for 8 weeks. Body weight, motor coordination, and muscle strength were monitored weekly from week 10 to week 18 (Figure 2A–C).

As previously reported in SOD1G93A mice [18], disease progression was paralleled by a continuing reduction in body weight (Figure 2A). Chronic RP administration did not affect the normal growth in WT mice and did not modify the loss of body weight in SOD1G93A transgenic mice (Figure 2A). Similarly, chronic administration of RP did not improve the loss of motor coordination in the rotarod test and the latency to fall was similar in vehicle (DMSO) or RP-treated SOD1G93A mice (Figure 2B).

Notably, loss of muscle strength of SOD1G93A mice in the hanging-wire test was significantly reduced in those receiving RP compared to those treated with DMSO (Figure 2C). To further support the functional meaning of this result, we checked for changes in the expression of collagen type XIX alpha 1 gene (*Col19α1*), a biomarker associated with ALS progression, whose expression in the muscle of SOD1G93A mice negatively correlates with longevity [20]. Quantitative PCR (qPCR) analysis of the quadriceps muscle from SOD1-DMSO mice showed a 6.5-fold increase in *Col19α1* mRNA expression compared to the WT-DMSO group (Figure 2D *Col19α1* mRNA levels in WT mice treated with RP were similar to those observed in the WT-DMSO group (Figure 2D). However, RP administration to SOD1G93A mice partially but significantly reversed the increase in *Col19α1* mRNA (Figure 2D). This result indicates a positive effect of the RP treatment to delay disease progression in the quadriceps muscle of SOD1G93A mice.

### 2.3. Treatment with RP Reduces Motoneuron Loss and Improves Gliosis in the Ventral Horn of SOD1G93A Spinal Cord

To further substantiate a neuroprotective effect of RP in the SOD1G93A mouse model of ALS, we next checked a potential rescue of ventral horn motoneuron loss. Nissl staining of lumbar spinal cord sections of WT and SOD1G93A mice showed, as reported [21], a significant reduction in the density of large soma motoneurons in SOD1-DMSO compared to WT-DMSO mice (Figure 3A,B). Interestingly, RP treatment partially, but significantly, reduced motoneuron loss in SOD1G93A mice compared to the SOD1-DMSO group. RP did not exert any effect on motoneuron density in the WT control groups (Figure 3A,B).

Moreover, we assessed whether RP was also affecting the glial reaction, a hallmark of neuroinflammation reported in SOD1G93A mice [22,23,24]. Immunohistochemical analysis of the ventral horn in the spinal cord showed that the expression of the microglial marker ionized calcium binding protein-1 (Iba1) was significantly higher in SOD1G93A mice than in WT mice (Figure 4A,B), suggesting the presence of activated microglia in the transgenic mice. Notably, RP-treated SOD1G93A mice presented a significant reduction in Iba1 staining compared to SOD1G93A mice treated with DMSO. The administration of repaglinide did not alter Iba1 expression in wild-type mice (Figure 4A,B). We also stained spinal cord sections with an antibody for the glial fibrillary acidic protein (GFAP), a marker expressed by astrocytes that is up-regulated in association with inflammatory processes in neurodegenerative diseases [25,26]. As previously reported in these mice [23,24], GFAP immunoreactivity was increased in the ventral horn of SOD1G93A transgenic mice compared to WT littermates (Figure 4C,D). RP treatment did not change GFAP expression in the WT mice compared to the WT-DMSO group. However, RP administration reduced GFAP immunoreactivity in SOD1G93A mice compared to the SOD1-DMSO group (Figure 4C,D).

Taken together, these results indicate that RP treatment was able to reduce neuron loss and gliosis in SOD1G93A transgenic mice without exerting any deleterious effect in WT animals.

### 2.4. RP Administration Stimulates ATF6 Processing in the Spinal Cord of SOD1G93A Mice

As mentioned, ATF6 is a key effector regulating the UPR in the endoplasmic reticulum (ER–UPR). In 2016, we reported that RP competes with the DREAM–ATF6 interaction, increasing ATF6 processing and promoting neuroprotection in a mouse model of Huntington’s disease [14]. To test whether a similar mechanism is involved in the neuroprotective effects exerted by RP in the spinal cord of SOD1G93A mice, we studied the effect of RP administration on ATF6 levels and its processing. Western blot analysis revealed increased total levels of ATF6, full-length (p90) plus processed form (p50) in SOD1G93A mice treated with vehicle compared to WT mice (Figure 5). Moreover, ATF6 processing, defined as the p50/p90 ratio, was not significantly different between SOD1G93A and WT-DMSO-treated mice (Figure 5). Interestingly, in SOD1G93A mice, RP administration increased the p50/p90 ratio and reduced total ATF6 (p90 + p50) levels compared to DMSO-treated SOD1G93A mice. These results linked the amelioration of disease symptoms following the administration of RP with the stimulation of ATF6 processing in the spinal cord of this ALS model. Finally, RP administration did not change total ATF6 levels nor ATF6 processing in WT mice (Figure 5).

## 3. Discussion

Accumulation of abnormal mutant SOD1 aggregates in the cytosol and the mitochondrial intermembrane space interferes with the assembly and maturation of cellular and mitochondrial proteins and triggers the unfolded protein response both at the endoplasmic reticulum (ER) and the mitochondria [15,27,28,29,30]. In addition, it has been suggested that mutant SOD1 protein interacts specifically with Derlin-1, a component of the ER-associated protein degradation (ERAD) machinery, and elicits ER stress-induced apoptosis in motoneurons through dysfunction of ERAD [31]. Several lines of evidence indicate that motoneurons in ALS are particularly sensitive to changes in ER stress [32,33]. Activation of the UPR at the ER (ER–UPR) depends on three ER transmembrane receptors that sense the ER stress signal; ATF6, inositol-requiring kinase 1 (IRE1), and double-stranded RNA-activated protein kinase-like endoplasmic reticulum kinase (PERK) [15]. In this study we have shown that chronic RP administration to SOD1G93A transgenic mice induces neuroprotection and activation of ATF6 processing. Notably, we showed that levels of total ATF6 are significantly increased in SOD1G93A spinal cords compared to wild-type mice, while the levels of activated ATF6 are similar in both genotypes. Previous in vitro studies have shown increased processing and nuclear translocation of ATF6 in N2a neuroblastoma cells overexpressing mutant SOD1G85R [34] and an increase in SOD1 aggregates in the motoneuronal cell line NSC-34 following ATF6 knockdown [35]. These data suggest a neuroprotective role of ATF6 activation in ALS. Our results support this notion, since blocking the DREAM–ATF6 interaction after RP administration in SOD1G93A mice significantly increased the content of the transcriptionally active processed form of ATF6 and the following activation of the ER–UPR was associated with a decrease in motoneuron loss and gliosis and to a reduction in disease markers. Downstream effectors of ATF6 activation and whether the activation of ATF6 processing is the only mechanism responsible for the neuroprotective action of RP are currently under investigation. Furthermore, not only ATF6 but also IRE1 and PERK, the other two branches of the ER–UPR, have been shown to be activated in different ALS experimental models as well as in patient samples [31,36,37,38,39]. Genetic manipulation of these two UPR branches confirmed their importance in ALS. Thus, conditional deletion or knock-down of abnormally high levels of XBP1, the main downstream effector of the IRE1 pathway, in SOD1 mutant mice or patients motoneurons, respectively, delayed disease onset and increased lifespan or conferred significant neuroprotection in vitro [35,40]. Similarly, genetic ablation of elevated levels of ATF4, the main downstream effector of the PERK pathway, in mutant SOD1G93A mice prolonged lifespan and slowed disease progression [41]. The involvement of the PERK pathway in ALS, however, has been challenged by a more recent study [42] showing that PERK haploinsufficiency has no effect on disease progression in five different lines of mutant SOD1 transgenic mice. Moreover, genetic ablation of CHOP, another transcriptional effector downstream in the PERK pathway, or induced deficiency of GADD34, which potentiates the PERK pathway by stabilizing phospho-eIF2α, did not change disease progression in the same five mutant SOD1 transgenic mouse lines [42].

Taken together, the available experimental evidence supports the functional significance of ER–UPR in the fate of ALS motoneurons. This triggered the development of many small molecules to specifically target the different UPR components and their assessment as feasible candidates for therapeutic intervention in ALS. Ideally, besides avoidance of unwanted side effects, a perfect candidate should potentiate the pro-survival function of the UPR, solving the protein aggregation problem and restoring protein homeostasis. At the same time, a perfect candidate should not over-stimulate UPR activity leading to neuronal demise. Several strategies specific for each UPR branch were developed and the results recently reviewed [43,44,45,46]. In brief, (i) to modulate the PERK pathway the strategies included the inhibition of PERK activation (GSK2606414, GSK2656157) [47,48], the block of downstream effects of elF2α phosphorylation (ISRIB, trazodone, dibenzoylmethane) [47,49,50,51,52] and the inhibition of elF2α phosphatases (guanabenz, salubrinal, sephin1) [33,53,54,55,56,57]; (ii) to modulate the IRE1 arm there are small molecules targeting the IRE1 RNase activity (4μ8C, salicylaldimines, SFT-083010, MKC-3946, toyocamycin) [58,59,60,61,62,63], to increase the XBP1 splicing levels (citrinins, Patulin, quercetin, apigenin) [64,65,66], and to block kinase activity and/or interfere IRE1 oligomerization (sunitinid, APY29, KIRAs) [67,68,69,70]; and (iii) to modulate ATF6, increasing or decreasing its expression (different flavonoids including fisetin, apigenin, luteolin, baicalein, kaempferol) [65,71,72]. Moreover, several pharmacological chaperons have been assessed to restore defective protein homeostasis in ALS, extensively reviewed in Tao and Conn (2018) [73], including arimoclomol [74], tauroursodeoxycholic acid (TUDCA) and sodium 4-phenyl butyrate (4-PBA) [75,76], and geldamycin [77,78]. As a result, these studies identified potential molecules targeting the UPR that could be useful for ALS treatment, and some of them have been already enrolled in clinical trials of the disease, e.g., guanabenz [54,79] and a combination of the chemical chaperones TUDCA and 4-PBA, currently running a phase III trial with 600 patients.

Accumulation of abnormal mutant SOD1 aggregates in the mitochondrial intermembrane space interferes with the assembly and maturation of mitochondrial proteins, triggering mitochondrial dysfunction, which ultimately activates the UPR at the mitochondria (MT–UPR) [27,29,30]. Like the ER–UPR, the MT–UPR is a transcriptional program that controls the expression of mitochondrial proteases and chaperones designed to restore protein homeostasis at the mitochondria [80,81]. Interestingly, induction of the MT–UPR in SOD1G93A spinal cord precedes the onset of disease symptoms, follows disease progression and, in some cases, decays at disease end-stage [82,83]. DREAM is located is several subcellular compartments, including the mitochondria. The role of DREAM in mitochondrial function has not been addressed, and whether RP administration has an effect on the mitochondrial response to SOD1G93A aggregation is currently unknown. The upcoming therapies to modulate the activation of mitochondrial UPR have been recently reviewed [84,85,86,87].

Chronic administration of RP has profound effects on motoneurons survival, the glial response, and muscle strength. Moreover, it reduces the levels of Col19α1 mRNA, a marker of ALS associated with bad prognosis and fast disease progression [20]. Nevertheless, RP did not change the onset of disease, rate of body weight loss nor the deficits in motor coordination in the rotarod test. Using a similar dosing and pattern of administration, RP delayed onset and slowed progression, including the amelioration of motor deficits in the R6/1 mouse model of Huntington’s disease. Nevertheless, like in this case in SOD1G93A mice, RP administration did not affect the loss of body weight in R6/1 mice. Whether an increase in the dosage or a change in administration pattern of RP could render a significant effect on motor coordination, body weight loss, or lifespan in ALS mice remains to be investigated. Moreover, the potential benefits of the new DREAM binding molecules of the PC-series, like PC330 or PC260 [14,88,89,90,91], in SOD1G93A mice remain to be investigated. Noteworthy, the SOD1G93A mouse model is a quite aggressive model of familial ALS, showing a very early onset and very fast disease progression [92]. Whether RP or the new PC-molecules could offer improved results in other mouse models of familial or sporadic ALS is currently under investigation.

## 4. Materials and Methods

### 4.1. Animals and Treatments

#### 4.1.1. Animals

Hemizygous B6SJL-Tg SOD1G93A males (stock number 002726) purchased from The Jackson Laboratory (Bar Harbor, ME, USA) were mated with C57BL/6J X SJL/J F1 hybrid females (B6SJLF1) purchased from Janvier Labs (Saint-Berthevin Cedex, France) to obtain hemizygous transgenic (SOD1G93A) and wild-type mice used in this study. The mice were housed at the animal facilities in Centro de Investigación Biomédica de Aragón (CIBA) under a standard light-dark (12:12) cycle. Food and water were provided ad libitum.

All procedures were approved by the Ethic Committee for Animal Experiments from the University of Zaragoza and in accordance with the Spanish Policy for Animal Protection RD53/2013, which meets the European Union Directive 2010/63 on the protection of animals used for experimental and other scientific purposes. The approval codes are PI45/22 and the approval date was 22 August 2022.

#### 4.1.2. Treatments

RP or vehicle DMSO were administered orally in the drinking water at a concentration of 2 μg/mL from week 10 to week 18 of age to SOD1G93A and wild-type littermates. Drinking water was renewed weekly. Groups were sex-balanced.

### 4.2. Weight Control and Behavioral Tests

Both WT and SOD1G93A, a total number of 62 mice, were weekly weighted from the beginning of the treatment at week 10 of age. In addition, rotarod and hanging-wire tests were performed weekly with the SOD1G93A mice, although the mice started to train on the rotarod one week before starting the treatment. Motor coordination was assessed using a rotarod (ROTA-ROD/RS, LE8200, LSILETICA, Scientific Instruments; Panlab, Barcelona, Spain). Animals were placed onto the cylinder at a constant speed of 14 rpm. The animals had three attempts to remain on the rotarod for a maximum of 180 s per trial, and the longest latency was recorded. Muscular strength was tested by the hanging-wire test. Each mouse was placed on a wire lid which was turned upside down. The latency from the beginning of the test until the mouse fell was timed. The animals had three attempts to remain for a maximum of 180 s per trial, and the longest latency was recorded. All behavioral experiments were carried out in blind conditions for treatment.

### 4.3. Western Blot

Eighteen-week-old WT and SOD1G93A mice treated with DMSO or RP, a total number of 77 animals, were sacrificed under deep anesthesia and their spinal cords were extracted. The spinal cord whole cell extracts preparation was performed as described [93]. In brief, tissue was homogenized on ice in RIPA buffer (9806, Cell Signaling Technology, Danvers, MA, USA) supplemented with protease inhibitor (Complete EDTA-free, Roche, Mannheim, Germany) and 1 mM PMSF (phenylmethanesulfonyl fluoride). Extracts were cleared by centrifugation (14,000× *g*, 20 min, 4 °C). Samples (20–30 μg) were analyzed by SDS-PAGE (sodium dodecyl sulfate-polyacrylamide gel electrophoresis) and immunoblot. PVDF (polyvinylidene difluoride) membranes were incubated (overnight, 4 °C) with specific antibodies to DREAM/KChIP3 (Ab730, [19]) and ATF6α (A303–719, Bethyl, Montgomery, TX, USA). Secondary antibodies used were HRP (horseradish peroxidase)-conjugated donkey anti-rabbit and IgG antibody (Jackson, Cambridgeshire, UK), and detection was with ECL Select (GE Healthcare, Buckinghamshire, UK). Equal protein loading was analyzed by Coomassie staining of the membrane after immunoblotting. Band and total lane (Coomassie) intensities were quantified with ImageLab 6.1 software (BioRad, Hercules, CA, USA). After densitometric analysis, band intensity ratio vs. Coomassie was normalized to DMSO-treated WT mice for each experiment.

### 4.4. Real-Time qPCR

RNA was isolated from the quadriceps muscle of 18-week-old WT and SOD1G93A mice treated with DMSO or RP, a total number of 26 animals, using TRIzol, treated with DNase (Ambion, Vilnius, Lithuania), and reverse transcribed using hexamer primer and Moloney murine leukemia virus reverse transcriptase. To confirm the absence of genomic DNA, each sample was processed in parallel without reverse transcriptase. An assay from Applied Biosystems (Mm00483576_m1, Vilnius, Lithuania) was used for the collagen type XIX alpha 1 gene (*Col19α1*) and the relative expression was quantified by the 2^−ΔΔCt^ method using hypoxanthine-phosphoribosyltransferase (*Hprt*) as a reference with primers 5′-TTGGATACAG GCCAGACTTTGTT-3′ and 5′-CTGAAGTACTCATTATAGTCAAGGGCATA-3′ and the MGB probe 5′-TTGAAATTCCAGACAAGTTT-3′.

### 4.5. Histology

#### 4.5.1. Tissue Processing

Eighteen-week-old WT and SOD1G93A mice treated with DMSO or RP, a total number of 22 animals, were sacrificed under deep anesthesia and spinal cords were extracted. Spinal cords were fixed with 4% paraformaldehyde (pH 7.4) for 48 h at 4 °C, cryoprotected in 30% sucrose (0.1 M PBS) and frozen, before sectioning into 20 µm-thick coronal sections on a cryostat.

#### 4.5.2. Immunofluorescence, Nissl Staining, and Image Analysis

Coronal spinal cord tissue sections were mounted on coated slides (Dako Flex IHC microscope slides), treated with sodium citrate 10 mM, pH 6.0, at 95 °C for 3 min, and preincubated with 5% normal goat serum (NGS) in phosphate-buffered saline (0.1 M PBS, pH 7.4)/0.1% Triton-X 100 for 60 min. The primary antibodies rabbit polyclonal anti-Iba1 (1:500; Wako Chemicals, Osaka, Japan) and mouse monoclonal anti-GFAP (1:250, Sigma-Aldrich, Saint Louis, MO, USA) were diluted in 0.5% NGS in PBS and applied overnight at RT. The secondary antibodies Alexa Fluor-568 goat anti-mouse IgG and Alexa Fluor-488 goat anti-rabbit IgG (1:500; both from Thermo Fisher, Rockford, IL, USA) were diluted in 0.5% NGS in PBS and applied for 1 h at RT. The slides were coverslipped in a medium containing p-phenylenediamine and bisbenzimide (Hoechst 33342; Sigma, Saint Louis, MO, USA). For Nissl technique, the sections were dehydrated in 1:1 alcohol/chloroform overnight, rehydrated, incubated with 0.1% cresyl violet solution for 5 min, rinsed in distilled water, dehydrated, cleared with xylene, and mounted with DPX (Fluka, Steinheim, Germany).

For quantification of fluorescence intensity, 8 or more high-resolution digital 200× widefield microphotographs from four consecutive sections per animal were taken at the same conditions of light and brightness/contrast, exposure time, and gain parameters, using a Leica DMI6000B epifluorescence microscope equipped with an OrcaR2 digital camera (Leica, Wetzlar, Germany). The microphotographs were used to measure the mean integrated density of labelling in the ventral horn of the spinal cord, using the ImageJ software (National Institutes of Health, Bethesda, MD, USA). Data were normalized to DMSO-treated WT mice for each experiment.

A Leica DM4 B microscope coupled with a Leica DMC 5400 digital camera (Leica, Wetzlar, Germany) were used to study and photograph the ventral horn of the spinal cords stained with the Nissl technique. High-resolution digital 100× widefield microphotographs (12 or more) from six consecutive sections per animal were taken at the same conditions of light and brightness/contrast, exposition time, and gain parameters. Polygonal shaped Nissl-positive cells showing a large soma (diameter ≥ 20 μm) resembling the morphology of large motoneurons were manually identified and counted as motoneurons within a ROI (region of interest) encompassing the ventral horn area. The diameter of large motoneurons was manually measured with the aid of the straight tool, and motoneurons were manually counted using the cell counter tool of ImageJ software (National Institutes of Health, Bethesda, MD, USA). ROIs encompassing the ventral horn were determined using the polygonal and ROI manager tools of the software. Cell counts were expressed as motoneurons/mm^2^.

### 4.6. Statistical Analyses

All data are shown as mean ± SEM. Differences were considered significant at *p* < 0.05. Parametric or nonparametric analysis was applied after normality test (Shapiro–Wilk). Student’s *t*-test was used to compare two groups. One-way or two-way ANOVA followed by multiple comparison between selected groups were performed for more than two groups. Prism GraphPad Software 8.0 (La Jolla, CA, USA) was used to plot graphs and for statistical analysis.

## 5. Conclusions

In conclusion, to target the defective proteostasis in ALS modulating, the ER–UPR seems like a plausible choice to develop new therapeutic venues. Importantly, this approach is transversal for different neurodegenerative diseases and successful molecules in the fine-tunning of the ER–UPR could be candidates, by themselves or in combination with other disease-specific therapies, for the treatment of these pathologies.

## Figures and Tables

**Figure 1 ijms-24-15783-f001:**
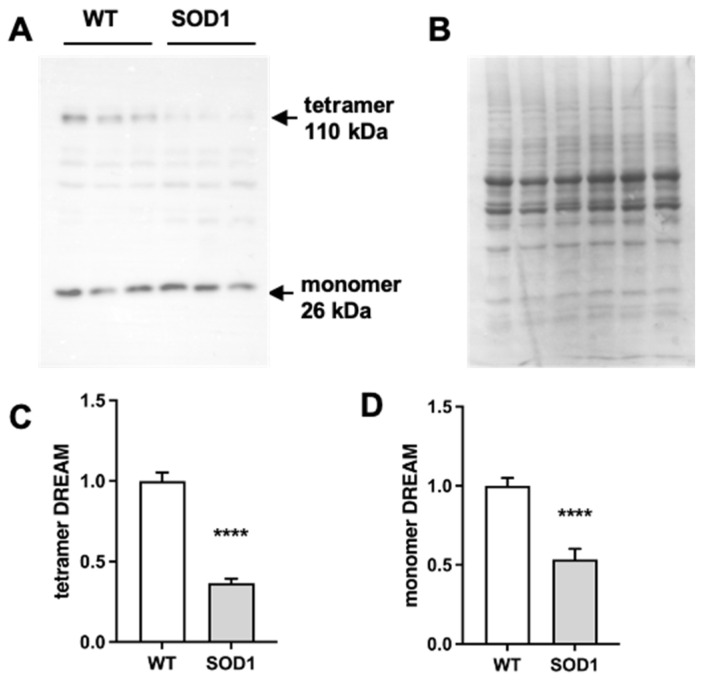
DREAM protein levels are reduced in the spinal cord of SOD1G93A mice. Western blot analysis of DREAM tetramer (**A**,**C**) and monomer (**A**,**D**) in spinal cord extracts from wild-type (WT) or SOD1G93A (SOD1). Representative blot for DREAM monomer and tetramer expression (**A**) and Coomassie staining (loading control) (**B**). The results represent the mean ± SEM of 11 WT and 12 SOD1 mice. **** *p* ≤ 0.0001 vs. WT (Student’s *t*-test).

**Figure 2 ijms-24-15783-f002:**
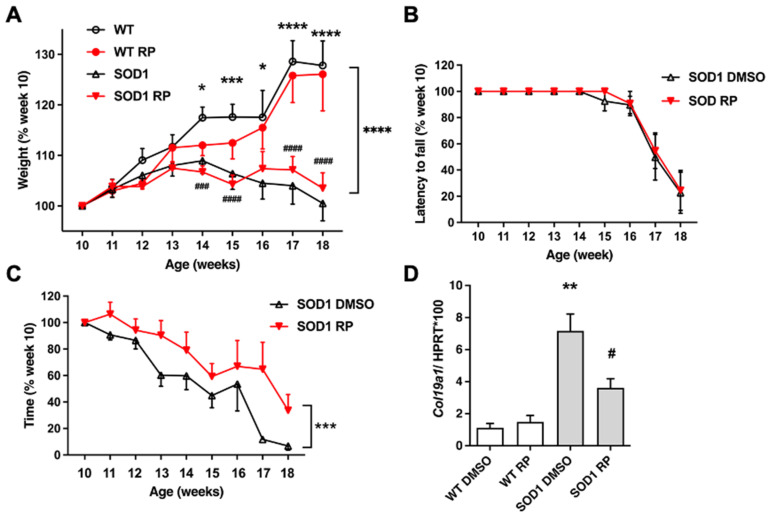
Repaglinide ameliorates motor strength loss in SOD1G93A mice. Weight, rotarod, and hanging-wire variation (%) were monitored from week 10 to 18 in wild-type (WT) and SOD1G93A (SOD1) mice treated with DMSO or repaglinide (RP). *Col19α1* expression was analyzed in the quadriceps muscle of WT and SOD1 mice treated with DMSO or RP. (**A**) Significant differences were found in the curves of weight variation between WT and SOD1 animals within each treatment group from week 14 onwards. Repaglinide did not protect from weight loss in SOD1 groups. (**B**) Similar values were obtained in the rotarod test between SOD1-DMSO and SOD1-RP groups. (**C**) Repaglinide improved latency in the hanging-wire test of SOD1 mice compared to SOD1-DMSO group. Significant differences were found between both curves. The results represent the mean ± SEM of 17 WT-DMSO, 14 WT-RP, 15 SOD1-DMSO, and 16 SOD1-RP mice in (**A**), 6 mice per group in (**B**) and 11 mice per group in (**C**). * *p* ≤ 0.05, *** *p* ≤ 0.001 and **** *p* ≤ 0.0001 vs SOD1-DMSO; ^###^
*p* ≤ 0.001 and ^####^ *p* ≤ 0.0001 vs. WT-RP in (**A**), *** *p* ≤ 0.001 vs. SOD1-DMSO in (**C**) (Two-way ANOVA + Holm–Šídák’s test). (**D**) Real-time qPCR analysis of *Col19α1* mRNA in the quadriceps muscle from WT and SOD1 mice receiving DMSO or RP. Values are normalized relative to Hprt mRNA levels. The results represent the mean ± SEM of 6 WT-DMSO, 6 WT-RP, 7 SOD1-DMSO, and 7 SOD1-RP mice. ** *p* ≤ 0.01 vs. WT-DMSO, ^#^ *p* ≤ 0.05 vs SOD1-DMSO, (One-way ANOVA followed by Holm–Šídák’s test).

**Figure 3 ijms-24-15783-f003:**
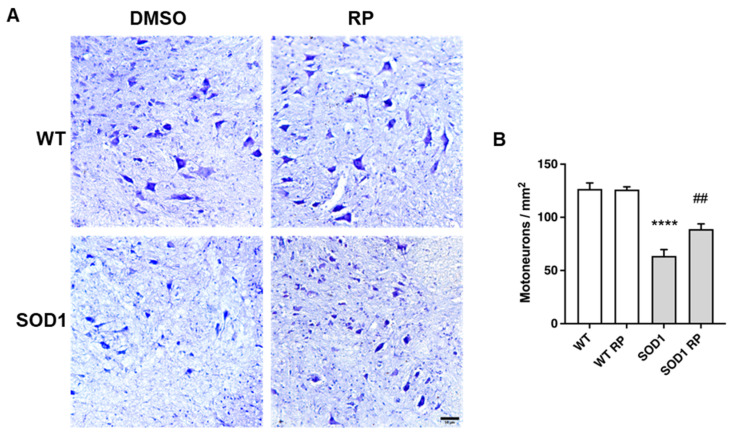
Repaglinide reduces the loss of motoneurons in the spinal cord of SOD1G93A mice. (**A**) Representative images of Nissl-stained motoneurons in the ventral horn of the spinal cord. (**B**) Quantification of large soma motoneurons density. The results represent the mean ± SEM of 5 WT-DMSO, 5 WT-RP, 6 SOD1-DMSO, and 6 SOD1-RP mice. **** *p* ≤ 0.0001 vs. WT-DMSO and ^##^ *p* ≤ 0.01 vs. SOD1-DMSO (One-way ANOVA followed by Holm-Šídák’s test). Scale bar, 50 µm.

**Figure 4 ijms-24-15783-f004:**
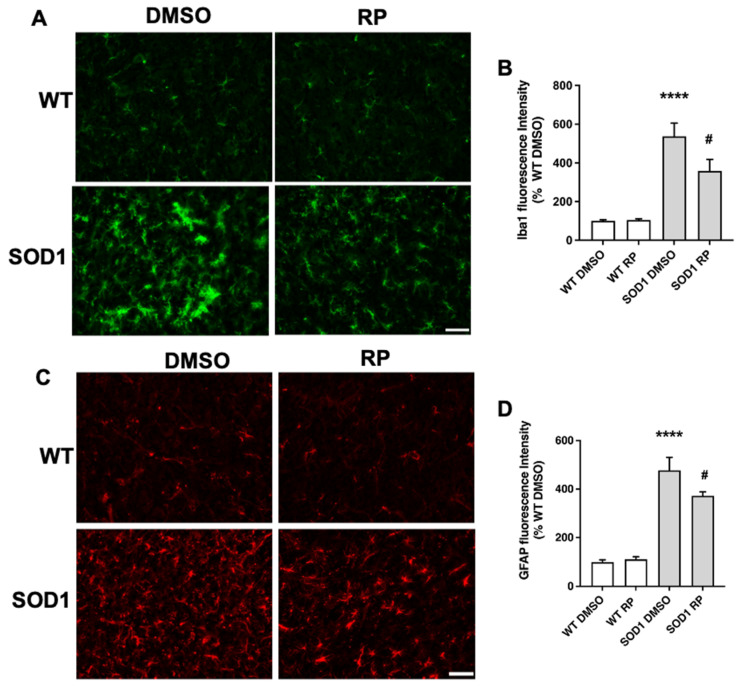
Repaglinide improves gliosis in the spinal cord of SOD1G93A mice. Representative images of microglial Iba1+ cells (**A**) and astroglial GFAP+ cells (**C**) in the ventral horn of the spinal cord. Fluorescence intensity quantification of Iba-1 (**B**) and GFAP (**D**) immunoreactivity. The results represent the mean ± SEM of 5 WT-DMSO, 5 WT-RP, 6 SOD1-DMSO, and 6 SOD1-RP. **** *p* ≤ 0.0001 vs. WT-DMSO and ^#^ *p* ≤ 0.05 vs. SOD1-DMSO (One-way ANOVA followed by Holm–Šídák’s test). Scale bar, 50 µm.

**Figure 5 ijms-24-15783-f005:**
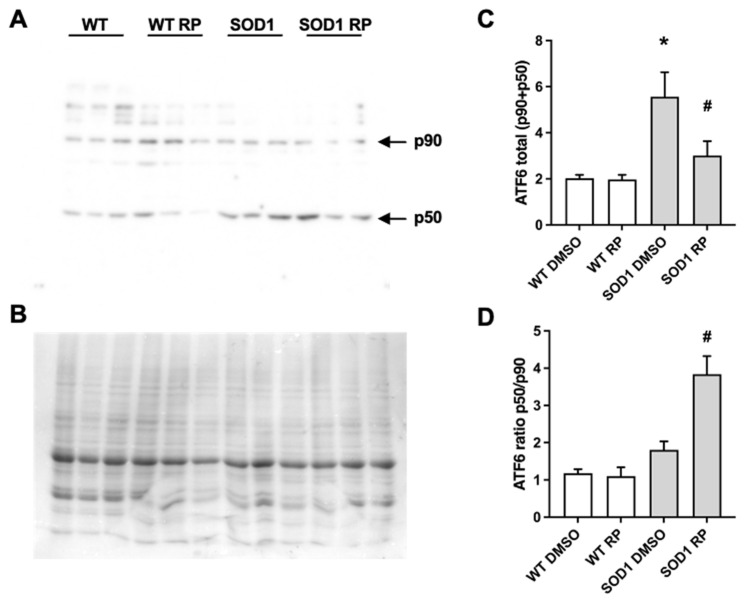
Repaglinide enhances activating transcription factor 6 (ATF6) processing in the spinal cord of SOD1G93A mouse. Western blot analysis of total ATF6 levels (full-length p90 and processed N-terminal p50) (**A**,**C**) and ATF6 processing (p50/p90 ratio) (**A**,**D**) in spinal cord of WT and SOD1G93A mice receiving DMSO or repaglinide. A representative blot is shown (**A**). Bands at 90 kDa and 50 kDa correspond to full-length ATF6 (p90) or processed N-terminal ATF6 (p50), respectively, and Coomassie staining (loading control) is shown (**B**). The results represent the mean ± SEM of 19 WT-DMSO, 18 WT-RP, 19 SOD1-DMSO, and 21 SOD1-RP mice analyzed by one-way ANOVA followed by Dunn’s test. * *p* ≤ 0.05 vs. WT-DMSO and ^#^ *p* ≤ 0.05 vs. SOD1-DMSO.

## Data Availability

All mentioned data are represented in the manuscript figures. Additional data will be made available upon request.

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
