# Peer review of "Repaglinide Induces ATF6 Processing and Neuroprotection in Transgenic SOD1G93A Mice"

_ijms, 2023, doi:10.3390/ijms242115783_

Round 1

Reviewer 1 Report

Comments and Suggestions for Authors

Gonzalo-Gobernado and colleagues investigated protective effects of repaglinide on ALS caused by mutant SOD1. Especially, repaglinide activated ATF6, one of ER stress pathways. Their findings are interesting; however, there are some serious concerns to accept these conclusions as described below.

1, Is the DREAM shown in Fig. 1A reduced lysates? If so, why does DREAM form a tetramer in Fig. 1A?

2, For WB loading control, it should be GAPDH or actin rather than CBB staining (Fig. 1 and 5).

3, Compared to cleaved ATF6, the signal of full-length ATF6 is weak. Is this really ATF6? It is thought that cleaved ATF6 is unstable due to its proteasome substrate. This induction of the cleaved form is thought to increase downstream factors such as GRP78. The authors need to investigate some targets of ATF6.

Author Response

Gonzalo-Gobernado and colleagues investigated protective effects of repaglinide on ALS caused by mutant SOD1. Especially, repaglinide activated ATF6, one of ER stress pathways. Their findings are interesting; however, there are some serious concerns to accept these conclusions as described below.

We acknowledge the comments from this reviewer. Changes in the text of the new version of the manuscript have been highlighted in red.

1, Is the DREAM shown in Fig. 1A reduced lysates? If so, why does DREAM form a tetramer in Fig. 1A?

Answer: Under our standard mild denaturing conditions and, specially, if we use non-reducing PAGE we always observe tetramer DREAM in addition to the monomer form in freshly prepared whole lysates from neural tissue or immune cells (Savignac et al., J. Immunol. 185:7527-36, 2010). In fact, the tetramer form is the only one that you detect if you prepare nuclear lysates (Savignac et al., J. Immunol. 185:7527-36, 2010) or you perform Southwestern blotting (Carrion et al. Mol. Cell. Biol. 18:6921-6929, 1998; Link et al., J. Neurosci. 24:5346-5355, 2004). Curiously, if you use recombinant DREAM protein, no matters the conditions you apply for the western blot, you will always detect monomer and dimer DREAM even under strong denaturing conditions (Palczewska et al., BBA Mol.Cell.Res. 1813:1050-8, 2011; Lopez-Hurtado et al., Sci Rep. 9:7260, 2019).

The tetrameric form of DREAM is the transcriptionally active configuration and as such it binds to DRE sites in the DNA in a calcium-dependent manner (Carrion et al. Nature 398:80-84, 1999).

2, For WB loading control, it should be GAPDH or actin rather than CBB staining (Fig. 1 and 5).

Answer: We respectfully disagree with this comment. In our experience, and it is a general belief, the use of GAPDH or actin for loading control in the case of samples from neural tissue from mouse models of neurodegenerative diseases is not the best option. The reason is that, quite often, the expression levels of these markers is not stable in disease conditions. Instead, Coomassie quantification does not present this problem and have many other advantages including a greatly improved linearity over immunodetection of GAPDH as discussed in Welinder and Ekblad J. Proteome Research 2010, 10:1416-1419.

3, Compared to cleaved ATF6, the signal of full-length ATF6 is weak. Is this really ATF6? It is thought that cleaved ATF6 is unstable due to its proteasome substrate. This induction of the cleaved form is thought to increase downstream factors such as GRP78. The authors need to investigate some targets of ATF6.

Answer: We have some experience working with different antibodies for ATF6 to detect full length and processed forms in different tissue samples as well as cultured cells (Naranjo et al., J Clin Invest. 126:627–638, 2016) and we are certain about those identified ATF6 bands. As in previous studies, in this case we compared the p50/p90 ratio in control and experimental samples included in the experiment and the results are shown in the figure.

We agree with the reviewer that to disclose downstream targets for ATF is very important. GRP78, also known as BiP, is one among a great number of potential downstream targets for ATF6 and we believe that there is no sense to try to identify one by one. So, in future experiments we plan to perform RNAseq to identify those that convey the initial activation of ATF6 processing to functional changes that could explain the amelioration of some symptoms in these mice after repaglinide treatment.

Reviewer 2 Report

Comments and Suggestions for Authors

The manuscript entitled ‘Repaglinide induces ATF6 processing and neuroprotection in transgenic SOD1G93A mice’ by Gonzalo-Gobernado et al. aims to demonstrate the potential therapeutic relevance of DREAM as a target for intervention in the context of ALS associated neurodegeneration. The subject of the study is original, but it has important problems in its fiction and structure and needs to be resolved.

Methods and results: groups and mouse numbers should be written clearly. In the text below the figures, it is stated that the groups included 5-6 mice for some parameters and 18 mice for others. This should be stated more clearly in the methods section. 5-6 mice in groups are statistically insufficient for histological analyzes. Groups should contain at least 9-10 mice. Therefore, this number should be increased. The number of motoneurons in the ventral horn was determined using ImageJ software under Nissl staining. There are several important issues here. First, there are different populations of motoneurons in the ventral horn that can be morphologically divided into large soma diameter and small soma diameter motoneurons. It is known that in neurodegenerative processes such as ALS, motor neurons with a large somatic slice are more affected, whereas those with a small somatic slice are less affected or respond better to treatment. In this study, the analysis did not distinguish between the different types of motor neurons, which may lead to misinterpretation. Second, ImageJ recognizes only the violet color difference that Nissl staining adds to the image. However, Nissl staining does not only stain a specific motor neuron, but also many non-specific regions. Therefore, specific analysis of Nissl staining with ImageJ is not possible. Moreover, after Nissl staining, somas and the nuclei they contain are used as markers under the microscope to determine whether the stained structures are motor neurons or not. ImageJ cannot make this distinction either. Therefore, if Nissl staining were to be used for motor neuron counting, more specific methods such as stereology would need to be chosen. It would be incorrect to claim that the number of motor neurons increases or decreases with the method used in MC.

Why was the 10th week chosen as the start of treatment? The 10th week corresponds to the symptomatic phase in these animals. If earlier stages, such as P30-40 (pre-symptomatic), had been chosen, the treatment's effectiveness might also have been different. Moreover, why was the experiment terminated at the 18th week? Does this time period coincide with the terminal phase? This should be clarified.

Discussion: Although the mice responded positively to the treatment, why did their body weights and motor function test results turn out negatively? An attempt should be made to explain this in detail in the discussion.

Author Response

The manuscript entitled ‘Repaglinide induces ATF6 processing and neuroprotection in transgenic SOD1G93A mice’ by Gonzalo-Gobernado et al. aims to demonstrate the potential therapeutic relevance of DREAM as a target for intervention in the context of ALS associated neurodegeneration. The subject of the study is original, but it has important problems in its fiction and structure and needs to be resolved.

Methods and results: groups and mouse numbers should be written clearly. In the text below the figures, it is stated that the groups included 5-6 mice for some parameters and 18 mice for others. This should be stated more clearly in the methods section. 5-6 mice in groups are statistically insufficient for histological analyzes. Groups should contain at least 9-10 mice. Therefore, this number should be increased.

We acknowledge the comments from this reviewer. Changes in the text of the new version of the manuscript have been highlighted in red.

Answer: As suggested by the reviewer, sample size has now been specified in the appropriate section of the material and methods section and in the figure legends.

Regarding to the minimum number of samples necessary for histological analysis, we respectfully disagree with referee 2. Below we provide few examples of papers published in well recognized Journals where the authors use similar sample sizes (n= 5-6 mice) for histological analysis in SOD1 mice models. Furthermore, in some of these studies they use large soma sizes and shape to identify affected motoneurons after using Nissl staining:

  • https://academic.oup.com/hmg/article/24/7/1883/595965 (Joyce et al. Human Molecular Genetics, Volume 24, Issue 7, 1 April 2015, Pages 1883–1897) Figure1, n=5
  • https://www.sciencedirect.com/science/article/pii/S0969996119303778 (Ghadge et al, Neurobiology of Disease Volume 136, March 2020, 104702) Materials and methods section 2.5, n=5.
  • https://genesdev.cshlp.org/content/22/11/1451.long (Nisitoh et al, Genes & Dev. 2008. 22: 1451-1464) Figure5, n=5.
  • https://www.cell.com/neuron/fulltext/S0896-6273(20)30191-4?_returnURL=https%3A%2F%2Flinkinghub.elsevier.com%2Fretrieve%2Fpii%2FS0896627320301914%3Fshowall%3Dtrue (Gerbino et al.,2020, Neuron 106,789–805) Figure 2, n=5

The number of motoneurons in the ventral horn was determined using ImageJ software under Nissl staining. There are several important issues here. First, there are different populations of motoneurons in the ventral horn that can be morphologically divided into large soma diameter and small soma diameter motoneurons. It is known that in neurodegenerative processes such as ALS, motor neurons with a large somatic slice are more affected, whereas those with a small somatic slice are less affected or respond better to treatment. In this study, the analysis did not distinguish between the different types of motor neurons, which may lead to misinterpretation.

Second, ImageJ recognizes only the violet color difference that Nissl staining adds to the image. However, Nissl staining does not only stain a specific motor neuron, but also many non-specific regions. Therefore, specific analysis of Nissl staining with ImageJ is not possible. Moreover, after Nissl staining, somas and the nuclei they contain are used as markers under the microscope to determine whether the stained structures are motor neurons or not. ImageJ cannot make this distinction either. Therefore, if Nissl staining were to be used for motor neuron counting, more specific methods such as stereology would need to be chosen. It would be incorrect to claim that the number of motor neurons increases or decreases with the method used in MC.

Answer: We appreciate the comments of the reviewer about the use of the ImageJ software to study Nissl-stained sections and we fully agree with them for the case of using the application in the automatic way. In our work, however, we manually identified and quantified motoneurons (diameter ≥ 20 um and polygonal shaped) with the aid of the cell counter tool. We apologize because in the previous version of the manuscript we made a mistake, instead of writing “cell counter plugin” in the material and method section we should have written “cell counter tool”. Therefore, we did not use an automatic plugin for cell counting, we only used the cell counter tool to manually mark and count motoneurons during the analysis, and this tool has no capabilities to perform automatic cell counts.

With respect to the size, shape and relative density of the large Nissl+ motoneurons that we were counting, we agree with the concerns of the reviewer and we have modified figure 3, now expressing cell counts relative to area in the new Fig. 3 in the revised manuscript.

Finally, we have included the following statement in the text to clarify the methodology and to prevent possible misinterpretations:

“Polygonal shaped Nissl-positive cells showing a large soma (diameter ≥ 20um) resembling the morphology of large motoneurons were manually identified and counted as motoneurons within a ROI (region of interest) encompassing the ventral horn area. The diameter of large motoneurons was manually measured with the aid of the straight tool and motoneurons were manually counted using the cell counter tool of ImageJ software. ROIs encompassing the ventral horn were determined using the polygonal and ROI manager tools of the software. Cell counts were expressed as Motoneurons / mm2

Why was the 10th week chosen as the start of treatment? The 10th week corresponds to the symptomatic phase in these animals. If earlier stages, such as P30-40 (pre-symptomatic), had been chosen, the treatment's effectiveness might also have been different.

Answer: As we already published (Olivan et al. Exp. Anim. 4(2):147-153, 2015), our colony of SOD1G93A mice does not show significant disease symptoms at 10 weeks of age. Furthermore, initiating therapy at an early symptomatic stage, as opposed to an asymptomatic one, enhances its potential for clinical applicability, as in patients the treatment would be administered upon the manifestation of symptoms.

Whether the treatment could had been more effective with an earlier administration is something to analyze in future studies in which we will check also the effect of new DREAM ligands in SOD1G93A mice (Lopez-Hurtado et al., Sci Rep. 9:7260, 2019; Peraza et al., Frontiers Mol. Neurosci. 12:11-18, 2019). Since this mouse model tries to resemble the familial form of this pathology, the option to advance the treatment prior to disease onset might have some significance.

Moreover, why was the experiment terminated at the 18th week? Does this time period coincide with the terminal phase? This should be clarified.

Answer: Yes, our colony of SOD1G93A mice enters in the terminal phase at week 18th after birth.

Discussion: Although the mice responded positively to the treatment, why did their body weights and motor function test results turn out negatively? An attempt should be made to explain this in detail in the discussion.

Answer: In our previous studies with a mouse model of Huntington’s disease (Naranjo et al., J Clin Invest. 2016;126(2):627–638), repaglinide offered a better, though transient, protection for motor symptoms but, like in this case, there was no significant effect on body weight loss in those mice. We believe that the progressive loss of body weight is the final consequence of many dysfunctions at the metabolic level. To try to explain the lack of effect of repaglinide in SOD1G93A mice escapes the goals of this investigation. A comment, acknowledging this lack of effect on body weight loss is now included in the discussion.

Round 2

Reviewer 2 Report

Comments and Suggestions for Authors

Changes in study design and sample numbers of some parameters would have improved the quality of MC. However, as far as I understand it, the authors cannot be said to view this positively. It is their own decision to present their work in this form. Therefore, the MC can be accepted in its current form.